# Membrane thickness, lipid phase and sterol type are determining factors in the permeability of membranes to small solutes

Jacopo Frallicciardi[1,3], Josef Melcr [2,3], Pareskevi Siginou[1], Siewert J. Marrink [2✉] & Bert Poolman [1✉]

Cell membranes provide a selective semi-permeable barrier to the passive transport of molecules. This property differs greatly between organisms. While the cytoplasmic membrane of bacterial cells is highly permeable for weak acids and glycerol, yeasts can maintain large concentration gradients. Here we show that such differences can arise from the physical state of the plasma membrane. By combining stopped-flow kinetic measurements with molecular dynamics simulations, we performed a systematic analysis of the permeability of a variety of small molecules through synthetic membranes of different lipid composition to obtain detailed molecular insight into the permeation mechanisms. While membrane thickness is an important parameter for the permeability through fluid membranes, the largest differences occur when the membranes transit from the liquid-disordered to liquid-ordered and/or to gel state, which is in agreement with previous work on passive diffusion of water. By comparing our results with in vivo measurements from yeast, we conclude that the yeast membrane exists in a highly ordered and rigid state, which is comparable to synthetic saturated DPPC-sterol membranes.

[1] Department of Biochemistry, University of Groningen, Nijenborgh 4, 9747 AG Groningen, the Netherlands. [2] Department of Biophysical Chemistry, University of Groningen, Nijenborgh 4, 9747 AG Groningen, the Netherlands. [3]These authors contributed equally: Jacopo Frallicciardi, Josef Melcr. ✉email: s.j.marrink@rug.nl; b.poolman@rug.nl

Membrane permeability is an essential property of cell membranes that regulates the passage of solutes and solvents into and out of cells or intracellular compartments. Permeation is either mediated by membrane transport proteins or channels or occurs via passive diffusion; here we focus on the latter. Biological membranes are relatively impermeable for most nutrients and ions, hence passive permeability contributes to the flux of a subset of molecules.

Passive diffusion is dependent on the molecular characteristics of both the solute and the lipid bilayer. While most literature focuses on the properties of the permeants, namely size, shape, and polarity[1–6], fewer studies tackle the effects of membrane lipid composition on passive diffusion. According to the solubility-diffusion model, the permeability coefficient ($P$) of a molecule is predicted to be inversely proportional to the thickness of the membrane, and proportional to the product of partition coefficient and diffusion coefficient in the membrane. As a proxy for the partitioning coefficient of a compound in the membrane, partitioning into organic solvents is traditionally considered. Indeed, a strong correlation of permeability with the octanol-water partition coefficient has been found in egg lecithin membranes, with the permeability varying over five to six orders of magnitude[7–9]. In numerous works, however, highly ordered membranes have been shown to exhibit permeability coefficients significantly different from those predicted by the solubility-diffusion model using the octanol-water partition coefficient for solubility[10–13]. Despite many studies on the passive permeability of biological membranes, there is no data that addresses systematically the role of lipid composition on the permeability of membranes for small molecules.

In our previous work[14,15], we developed an assay and analysis method to accurately determine the permeability coefficient of synthetic membranes and the plasma membrane of living cells for small molecules such as weak acids and bases, water, and neutral solutes. The vesicles or cells are osmotically shocked by the permeant and the rate of volume recovery or cytosolic acidification/alkalinization is analyzed. By testing vesicles of different lipid composition, we observed a relationship between acyl chain saturation and membrane permeability for both water and formic and lactic acid.

Here, we combine the experimental work with coarse-grained (CG) molecular dynamics (MD) simulations to unravel the determinants of small molecule permeability of lipid membranes. MD simulations have previously revealed that the permeability of small molecules depends not only on their chemical nature, but also on the membrane properties[5,9,16,17]. For instance, the permeability of water molecules through model liquid-ordered ($L_o$) membranes is lower than through model liquid-disordered ($L_d$) membranes[18]. Moreover, as is also shown in[5], this study reveals why the solubility diffusion model breaks down: it does not capture the differences between the two membranes well when the compound solubility is assumed to be the same in the two membrane environments as in the commonly used octanol. This difference becomes increasingly important in membranes with more complicated compositions and rich phase behavior like the plasma membrane of yeast, which contains not only patches of highly ordered lipids in the $L_o$ phase, but also regions in the gel ($L_\beta$) phase[19–22]. Such details can be obtained from MD simulations, in particular using a CG model which enable systematic exploration of membrane composition and state points[23,24], allowing us to find the main descriptors of the permeability through biomembranes.

Using the combined approach of MD simulations and experimental fluorescence measurements performed at 20 °C we determine the permeability coefficients of several polar compounds (weak acids and glycerol) through membranes of different lipid compositions. The compositions span varying saturation levels, acyl chain lengths and sterol concentrations with a special emphasis on membranes in different physical states. By comparing the experiments with the results from MD simulations, we are able to interpret the observed changes in terms of solubility and solvent proximity. Finally, we compare the resulting permeability coefficients with the values we estimated in vivo for *Saccharomyces cerevisiae*[14] and use the permeability measurements as indirect reporters of the physical state of the yeast plasma membrane.

## Results

**Permeability and partitioning coefficients from experiments and simulations.** In experiments, we determine the permeability coefficients using a fluorescence-based assay that reports volume changes of vesicles by means of calcein self-quenching fluorescence[14]. Briefly, we follow the out-of-equilibrium relaxation kinetics of vesicles upon osmotic upshift with a stopped-flow apparatus, by addition of an osmolyte to the vesicle solution. The thermodynamic equilibrium is re-established by the flux of water and/or the osmolyte. The contribution of the two fluxes to the recovery kinetics depends on the relative permeability of water and the osmolyte. The permeability coefficients are then calculated using the previously developed technique and analysis tool[14,15]. Measured vesicle volume relaxation curves from solutes that permeate with slow or fast kinetics (formic acid, L-lactic acid, glycerol), and from non-permeating solutes like KCl, are shown in Fig. 1A. In the case of the weak acids, the vesicle volume recovers only partially because the acids were added as sodium salts, and sodium ions do not permeate through lipid membranes on the timescale of the measurements. In contrast, glycerol leads to an overall inflation of the vesicles above their original volume after the concentrations in/out of the vesicles are equilibrated. This is caused by the preferential partitioning of glycerol to the membrane-water interface and its interactions with lipids.

In MD simulations, we have used the coarse-grained (CG) Martini 3 model[25] to describe the permeation of a range of hydrophobic and hydrophilic generic solutes at infinite dilution, using the inhomogeneous solubility-diffusion model in which the permeation rate is obtained as an integral across the membrane of the ratio between local solubility and friction (see "Methods", Eq. 2). Briefly, we obtain free energy profiles of the solutes as a function of the distance from the membrane center (linked to the local solubility via the Boltzmann factor) by sampling its translocation through the membrane as schematically described in Fig. 1B. The solutes are modeled as single CG beads and labeled according to their hydrophilicity with levels I–IX. The most hydrophobic particles, levels VIII and IX, have $\log P_{OW}$ between butyric and sorbic acids, level V represents the hydrophobicity of pyruvic acid, and the most hydrophilic particle, level I, has $\log P_{OW}$ comparable to phosphoric acid. The free energy profiles through a DOPC membrane of solutes with different logarithm of octanol-water partition coefficients, $\log P_{OW}$, and their solvent accessibility are presented in Fig. 1C and D, respectively. Profiles of local friction are in Supplementary Fig. 1.

We compare both methods by plotting the dependence of the obtained permeability coefficients, $P$, on the logarithm of the octanol-water partitioning coefficients, $\log P_{OW}$, of several hydrophobic and hydrophilic solutes (eight weak acids and glycerol) in Fig. 1E, F, and G (numerical values in Supplementary Table 1). The permeability coefficients from MD simulations are somewhat larger than in experiments because of the intrinsically faster dynamics in coarse-grained modeling, but the relative changes of the permeability coefficients with $\log P_{OW}$ are in good accordance with experiments. For instance, the slope of the dependence of $P$ on

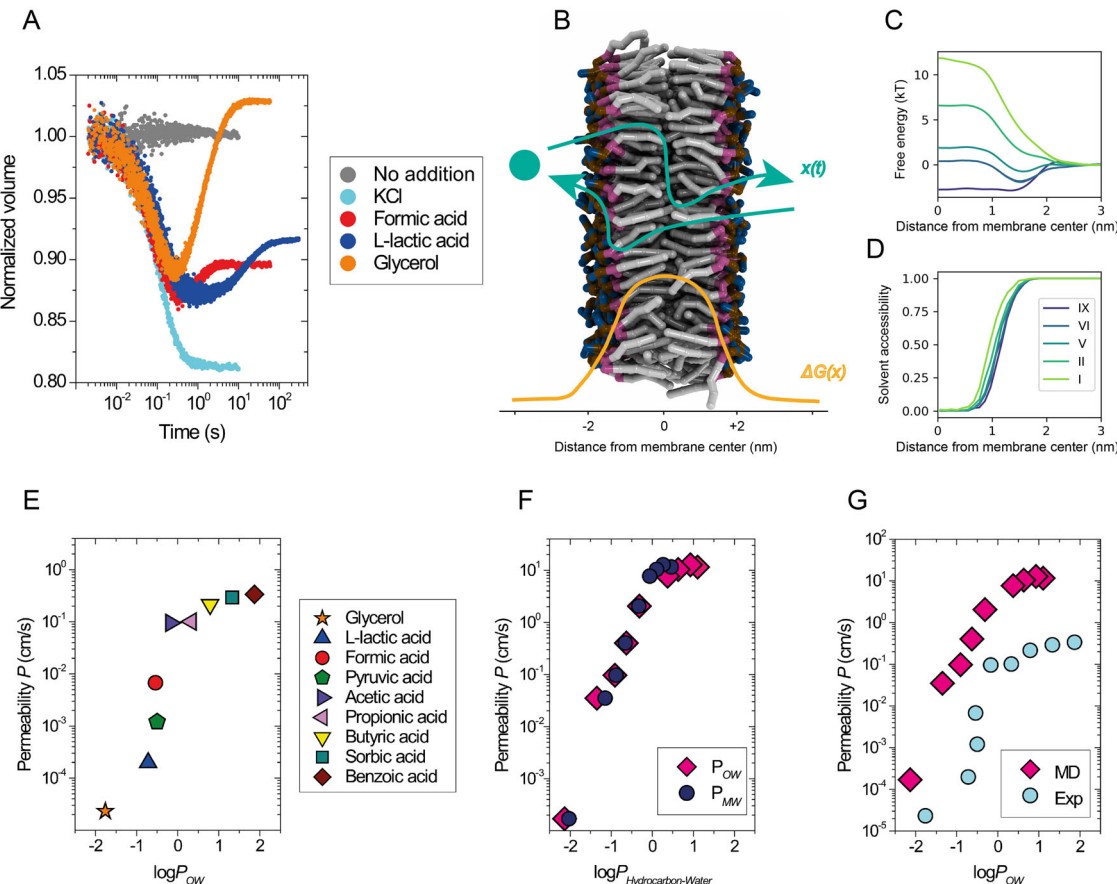

**Fig. 1 Permeability of solutes as a function of their lipophilicity. A** Overview of experimental assay. Kinetic data obtained with the calcein self-quenching assay using vesicles composed of DOPC mixed with buffer (gray) or osmotically shocked with 52.5 mM KCl (cyan), 50 mM sodium formate (red), 50 mM sodium L-lactate (blue) or 120 mM glycerol (orange) at 20 °C. **B** Schematic description of the permeation process, $x(t)$, through a lipid membrane with an example free energy profile, $\Delta G(x)$ (lipid tails, gray; glycerol moiety, purple; phosphate moiety, ochre; and choline moiety, blue; water molecules are not shown). **C** Selected free energy profiles from simulations of solutes with varying hydrophobicity levels (I most hydrophilic, IX most hydrophobic) permeating through a DOPC lipid membrane as a function of the distance from the bilayer center along the membrane. Only one half of the whole symmetric permeation profile is shown. **D** Solvent accessibility profiles of the permeating solutes along the permeation pathway. Solutes interact with solvent molecules even deep in the membrane tail region ($x < 1.0$ nm). **E–G** Permeability coefficients for DOPC membranes from experiments at 20 °C (**E**, **G**) and simulations (**F**, **G**) plotted against the logarithm of octanol/water ($\log P_{OW}$) and membrane/water partitioning coefficient ($\log P_{MW}$) of the solutes. Solutes of the hydrophobic levels I ($\log P_{OW} = -2.14$) to IX ($\log P_{OW} = 1.1$) were used in the simulations. $\log P$ is $\log_{10}$ (partition coefficient), where $P$ refers to the equilibrium distribution of a molecule between the hydrophobic and hydrophilic phase of two immiscible solvents. The experimental $\log P_{OW}$ values for weak acids and glycerol (**E**, **G**) were taken from the PubChem database (https://pubchem.ncbi.nlm.nih.gov/). The simulated $\log P_{OW}$ values (**F**, **G**) were taken from work[25]; $\log P_{MW}$ values (**F**) were obtained from the present simulations. The partition coefficients are related to the free energy difference between the respective phases. In the case of $P_{MW}$, this is the difference between the membrane center and the solvent phase. The values of the free energy are presented in (**C**). The numerical values are presented in Supplementary Table 1.

$\log P_{OW}$ for the hydrophobic and hydrophilic solutes is independently recovered in both methods. In particular, the difference between acetic acid ($\log P_{OW} = -0.17$) and formic acid ($\log P_{OW} = -0.54$) corresponds to approximately one order of magnitude change of their permeability from experiments, whereas a similar difference from propionic acid ($\log P_{OW} = 0.33$) to butyric acid ($\log P_{OW} = 0.79$) is reflected only by about a factor of 2 increase of the permeability. Interestingly, for the hydrophobic compounds $\log P_{MW}$ from MD simulations is smaller than $\log P_{OW}$ (Fig. 1F). This is mainly caused by non-vanishing interactions of the solutes with solvent molecules even below 1 nm from the membrane center (Fig. 1D), making the overall polarity of the lipid membrane interior generally higher than in bulk octanol.

**Phospholipids tail length affects permeability by changing the membrane thickness.** We determined permeability coefficients through phospholipid bilayers of different thickness with monounsaturated tails of lengths between 14 and 26 carbons (Fig. 2A experiments and 2B simulations). The permeability coefficients for formic acid, L-lactic acid and water from experiments decrease with increasing chain length (Fig. 2A), and they are in line with the calculated values from simulations using the solute of hydrophobic level III as a representative for a generic polar weak acid in neutral form (Fig. 2B). Increasing the length of the phospholipid acyl tails increases the hydrophobic thickness of the membrane (Fig. 2B and C, and Supplementary Fig. 2). Elongating the lipid tails by 2 carbons decreases the permeability coefficients by approximately a factor of 1.5 for tail lengths from 14 to 26 carbons.

A higher membrane thickness leads to a net shift in the position of the membrane interface with water (Fig. 2C bottom). The free energy barrier shows corresponding net shifts of the profiles, yielding wider barriers for thicker membranes (Fig. 2C

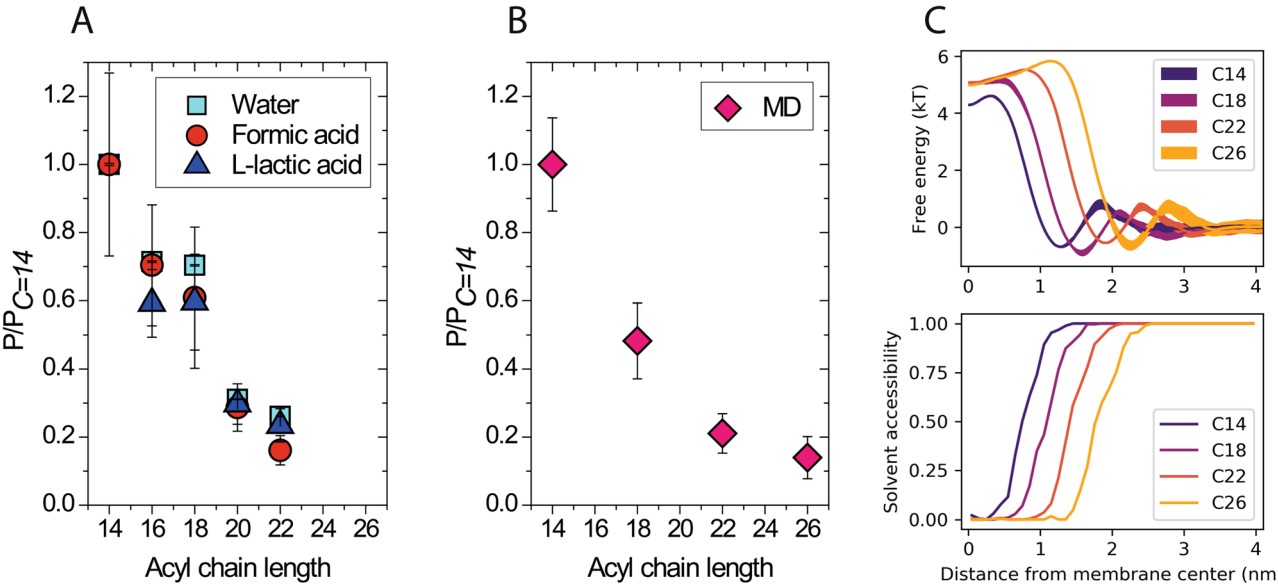

**Fig. 2 Permeability of solutes as a function of acyl chain length.** Permeability coefficients as a function of acyl tail length from experiments at 20 °C (**A**) and simulations (**B**). Lipids with mono-unsaturated tails of lengths between 14 and 26 carbons correspond to 1,2-dimyristoleoyl-*sn*-glycero-3-phosphocholine (C = 14), 1,2-dipalmitoleoyl-*sn*-glycero-3-phosphocholine (C = 16), 1,2-dioleoyl-*sn*-glycero-3-phosphocholine (C = 18), 1,2-dieicosenoyl-*sn*-glycero-3-phosphocholine (C = 20), 1,2-dierucoyl-*sn*-glycero-3-phosphocholine (C = 22) and 1,2-dihexacosenoyl-*sn*-glycero-3-phosphocholine (C = 26), respectively. The level of unsaturation is kept constant (mono-unsaturated tails) for all the analyzed lipids. Solute of the hydrophobic level I was used in the simulations. Permeability coefficients are normalized to $P_{C=14}$, which in the experimental analysis corresponds to 22.8 (±1.9) × $10^{-3}$ cm/s, 11.80 (±1.43) × $10^{-3}$ cm/s, and 0.332 (±0.052) × $10^{-3}$ cm/s for water, formic acid, and L-lactic acid, respectively. Data are presented as mean values ± SEM. Error estimates represented by error bars around the mean values are described in "Methods" ("Fit of the in vitro kinetics" for experimental data, "Inhomogeneous solubility-diffusion model", for MD simulations). The numerical values are presented in Supplementary Table 2. **C** Free energy (top) and solvent accessibility (bottom) profiles from simulated membranes of variable thickness. The uncertainty of the free energy profiles is represented by the thickness of the lines. Longer acyl chain length of the phospholipid tails increases the membrane hydrophobic thickness (Supplementary Fig. 2), which leads to an increasingly wider and higher free energy profile.

top). Moreover, the free energy barriers reveal a decreasing height for thinner membranes. This is linked both to the overall increased polarity of the membrane interior for thin bilayers with respect to thick bilayers, as the solvent molecules are closer to the membrane center for very thin bilayers (i.e., C14 in Fig. 2C), and to the changes of the lipid packing as reflected in the changes of average lipid surface areas (Supplementary Fig. 2).

**Membrane phase transition from $L_d$ to $L_\beta$ decreases the permeability of the membrane by orders of magnitude.** We have measured permeability coefficients of water, L-lactic acid, formic acid, and glycerol through lipid membranes with different degrees of unsaturation (d); we also measured glycerol as the plasma membranes of examined bacteria and yeast have orders of magnitude difference in permeability for this osmolyte. Mixtures of saturated (DPPC) and unsaturated lipids (POPC and or DOPC) have been studied with different respective fractions. We define the degree of unsaturation, d, as the ratio between the number of lipid tails with carbon-to-carbon double bonds ($N_{C=C}$) and the total number of tails ($N_{total}$): $d = N_{C=C}/N_{total}$. All the phospholipids used in this work have two tails per head group and up to one double bond per tail. Pure mixtures of DOPC, POPC and DPPC have degrees of unsaturation of 1, 0.5, and 0, respectively. d values of 0.84 and 0.67 were obtained by mixing DOPC and POPC in a 68:32 and in a 34:66 ratio, respectively. d values of 0.34 and 0.17 were obtained by mixing POPC and DPPC in a 68:32 and in a 34:66 ratio, respectively.

From both our measurements and MD simulations, we observe that decreasing the degree of unsaturation of the phospholipid acyl chains from 1 to 0.17 leads to a corresponding gradual

decrease of the permeability coefficient for all the compounds (Fig. 3A and B). This is in agreement with previous observations regarding water permeability by Olbrich and colleagues[26]. The decrease of the permeability coefficients arises from wider and higher free energy profiles as shown in Fig. 3C. In line with previous work[14], the effects of the degree of unsaturation on the permeability coefficient are almost independent of the chemical nature of the permeants in the range of d between 1.0 and 0.17 (Fig. 3A).

Importantly, for membranes with a degree of unsaturation below d = 0.17, we observe very dramatic changes of the permeability coefficients. Namely, the permeability coefficient through DOPC (d = 1.0) versus DPPC (d = 0.0) membranes decreases approximately 200-fold for water, and 2000-fold for formic acid and glycerol (Fig. 3A experiments and 3B simulations). Moreover, permeation of lactic acid was not observed in the experiments with DPPC vesicles on the timescale of 10 h.

The enormous leap in the permeability coefficient arises mainly from the highly decreased solubility of the compounds in the membranes with low degrees of unsaturation as seen in Fig. 3C. The free energy profiles from MD simulations show not only a wider but also much higher barriers, which form the major contribution to the decrease of the permeability coefficients. In addition, the mobility of the permeating solutes is significantly decreased slowing the permeation even further, however, this effect is lower for smaller solutes (Supplementary Fig. 3).

The sudden non-smooth changes of membrane properties, including the differences in permeability coefficient, are directly linked to the phase state of the membranes, which is in line with previous studies on water and oxygen permeability in membranes of different phase state[18,26,27]. In Fig. 3B, we plot the calculated

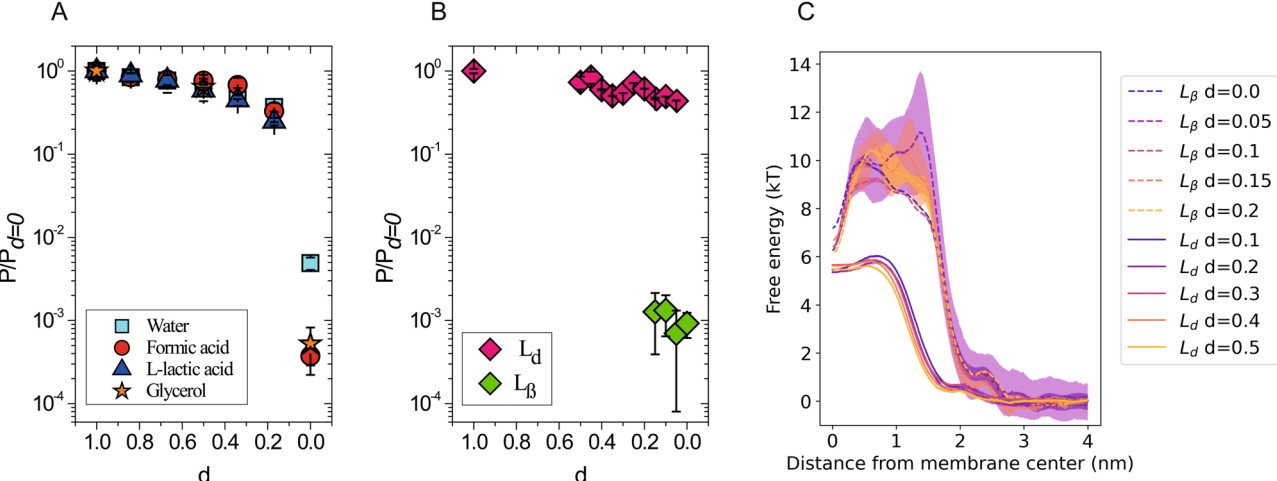

**Fig. 3 Permeability of solutes as a function of acyl tail unsaturation.** The permeability coefficients for each solute are normalized to their respective value at $d = 1.0$, which in the experimental analysis corresponds to 16.0 (±1.7) × $10^{-3}$ cm/s, 6.73 (±0.74) × $10^{-3}$ cm/s, 0.198 (±0.033) × $10^{-3}$ cm/s, and 2.3 (±0.2) × $10^{-6}$ cm/s for water, formic acid, L-lactic acid, and glycerol, respectively. Permeability coefficients at 20 °C for water, formic acid, L-lactic acid, glycerol from experiments (**A**) and permeability coefficients for the solute of the hydrophobic level I from MD simulations (**B**) for membranes in the $L_d$ and $L_\beta$ phase. Simulations in the range of $d$ between 0.17 and 0.05 show results from both membrane phases, $L_d$ and $L_\beta$, which were used as a starting configuration and remained meta-stable within the simulation time. Data are presented as mean values ± SEM. Data are presented as mean values ± SEM. Error estimates represented by error bars around the mean values are described in "Methods" ("Fit of the in vitro kinetics" for experimental data, "Inhomogeneous solubility-diffusion model", for MD simulations). The numerical values are presented in Supplementary Table 3. **C** Free energy profiles from simulated membranes at $L_d$ phase (solid lines) and $L_\beta$ phase (dashed) with varying degree of unsaturation $d$.

permeability coefficients from simulations for different phases, that are, $L_d$ and $L_\beta$. The membrane phase state remains *meta-stable* on the simulation time scales, giving rise to two distinct values for the permeability coefficient in the region of $d$ between 0.15 and 0.05. While the free energy and friction profiles from the membrane at the $L_d$ phase compare well to those of the POPC membrane ($L_d$ phase under the same conditions), the profiles from membranes at the $L_\beta$ phase are similar to those of DPPC membranes ($L_\beta$ phase) in Fig. 3C.

We corroborate our findings by DSC measurements performed in this work (Supplementary Fig. 4) and found in literature[28], which show a decreasing membrane melting temperature with increasing degree of unsaturation. In particular, the POPC/DPPC mixtures with $d = 0.17$ and $d = 0.34$ form a stable $L_d$ phase with an interface to the $L_\beta$ phase through possible coexistence (Supplementary Fig. 4A); note the broad transition peaks around 24 °C ($d = 0.34$) and 34 °C ($d = 0.17$). Phase coexistence has been previously reported in GUVs prepared from the same lipid species by fluorescence measurements using probe partitioning[29].

**Sterols modulate the solute permeability by affecting the membrane phase state.** We have selected cholesterol and ergosterol as the main sterols of mammalian and yeast plasma membranes[30,31], and assessed their effects on the membrane permeability at molar levels from 0 to 45%. Permeability coefficients of membranes with varying concentrations of sterols and unsaturation index from experiments and simulations are shown in Fig. 4, and all numerical values are given in Supplementary Table 4.

For the membranes with POPC ($d = 0.5$), both ergosterol and cholesterol lead to a small but significant decrease in permeability. In line with the known smaller condensing effect of ergosterol compared to cholesterol on lipid bilayer structure[32,33], the decrease in permeability coefficient upon adding 45 mol% sterols is only approximately 2-fold for ergosterol but 5-fold for cholesterol. Similar effects are observed when titrating both sterols in vesicles composed of DOPC (Supplementary Table 4

and Supplementary Fig. 5) and have been previously reported for water permeability when introducing cholesterol in giant vesicles of DOPC or SOPC[34]. Our simulations show that in analogy to a decreasing unsaturation index in membranes without sterols (Fig. 3), the permeability coefficient decreases with increasing sterol concentration in POPC vesicles because of higher and wider free energy barriers (Fig. 4D). The effects of adding cholesterol are smaller in simulations than in the experiments but they are in line with the general trend.

The phase change from $L_d$ to $L_o$ in the simulations with POPC and between 15% and 30% sterol leads to a notable shoulder in the profile around 1.0 nm distance from the membrane center, where it is accompanied by a small depression in the profile at 0.0 nm (Fig. 4D). The corresponding changes of the permeability coefficients and free energy profiles are in accordance with our experiments (Fig. 4 and Supplementary Fig. 5) as well as with the recent simulations that compare the permeability of water through membranes in $L_o$ and $L_d$ phases[18]. The characteristic features of the free energy profiles of the $L_o$ phase are linked to the sterol-induced changes in lipid packing[35] and changes in the lateral pressure profile[36]. Unless a phase transition occurs, the shoulder in the profile continues to smoothly build up with decreasing unsaturation index towards DPPC (i.e., towards $d = 0.0$; Supplementary Fig. 6).

Titration of sterols into fully saturated DPPC bilayers ($d = 0.0$) has the opposite effect on solute permeability than in POPC membranes. For instance, we observe an overall increase of the permeability coefficient of about one order of magnitude between DPPC membranes in the absence or presence of 45 mol% of cholesterol or ergosterol. This large difference is caused by the change of the membrane phase from $L_\beta$ to $L_o$ upon addition of sterols. Opposite to what is seen in POPC membranes, sterols perturb the highly ordered acyl chains of the DPPC lipids leading to a significant decrease of the friction coefficient, e.g., compare DPPC bilayers to DPPC plus 15% cholesterol (Supplementary Fig. 3). As the free energy profiles for the DPPC membrane with 0 and 15 mol% of cholesterol are comparable in height and width, the change in the friction through the membranes in the $L_\beta$ phase

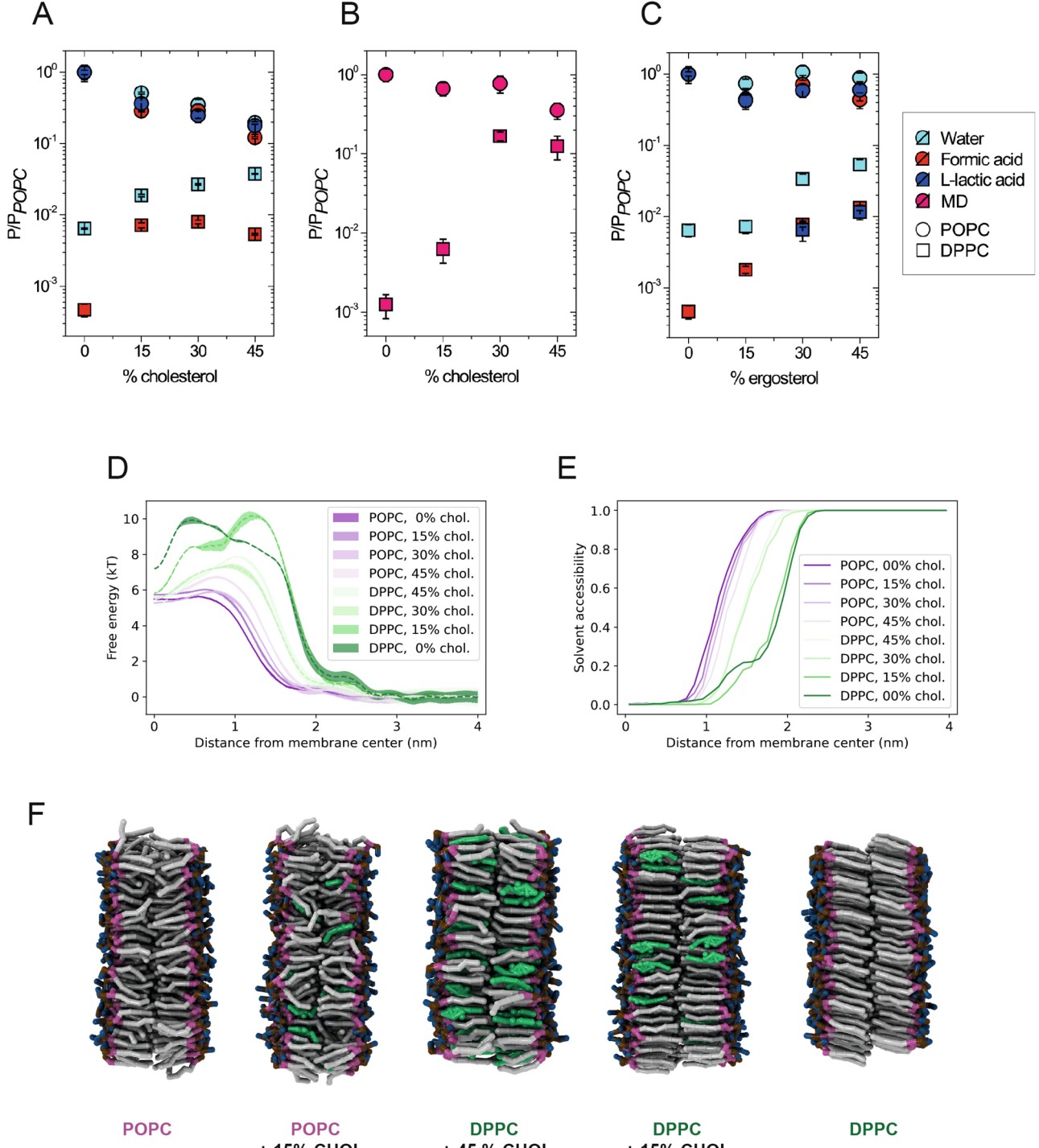

**Fig. 4 Permeability of solutes as a function of sterol concentration.** Permeability coefficients as a function of cholesterol (**A**, **B**) and ergosterol (**C**) concentration in POPC or DPPC vesicles from experiments (**A**, **C**) and simulations (**B**) at 20 °C. The permeability coefficients for each solute are normalized to the value in POPC vesicles, i.e., without sterol present; the permeability coefficient in POPC vesicles was 12.0 (±1.4) × 10$^{-3}$ cm/s, 5.53 ± (0.74) × 10$^{-3}$ cm/s, and 0.117 (±0.022) × 10$^{-3}$ cm/s for water, formic acid, and lactic acid, respectively. Solute of the hydrophobic level I was used in the simulations. Data are presented as mean values + SEM. Data are presented as mean values ± SEM. Error estimates represented by error bars around the mean values are described in "Methods" ("Fit of the in vitro kinetics" for experimental data, "Inhomogeneous solubility-diffusion model", for MD simulations). The numerical values are presented in Supplementary Table 4. **D**, **E** Free energy (**D**) and solvent accessibility (**E**) profiles from simulated membranes in the L$_d$ phase and L$_\beta$ or L$_o$ phase at varying cholesterol concentrations. The largest changes in the profiles and in the permeability coefficients are between simulations in different phase states. **F** MD simulation snapshots of lipid membranes of various compositions at different phase states (lipid tails gray; glycerol moiety, purple; choline moiety, blue; phosphate moiety, ochre; cholesterol, green; water molecules are not shown).

is an important factor for the difference in the permeability coefficient between the two lipid compositions.

Interestingly, the added cholesterol prevents water molecules from following the permeating solute into the membrane hydrophobic core. While water molecules can follow the permeating solutes into the DPPC membrane even below 1.0 nm (Fig. 4E), adding 15% cholesterol to the membrane keeps the water molecules above that threshold. This is linked to the shift of the free energy barrier peak from 0.5 nm in DPPC to 1.3 nm in DPPC plus 15% cholesterol and indicates that the sterols enforce solute dewetting further away from the hydrophobic region of the membrane. This effect comes from the highly anisotropic structure of the DPPC bilayer in the $L_\beta$ phase, which may form "wedge-like" defects to accommodate the permeating solutes through its highly ordered structure and expose an accessible surface for the solvent molecules to enter "for free"; these cavities are non-existent when sterols are present in the membrane.

Adding more than 15% sterols to DPPC membranes leads to a change in the membrane phase from $L_\beta$ to $L_o$ and another order of magnitude increase in the permeability coefficient, which is a direct consequence of the dramatically lowered free energy barrier (Fig. 4D). Increasing cholesterol from 30% to 45% in membranes in the $L_o$ phase leads to a decrease of the permeability coefficient, similar to what is observed for membranes with POPC. As the compositions with more than 30% cholesterol are in the same $L_o$ phase, this effect can be attributed to the condensing effect of cholesterol on lipid bilayer structure[32,33]. Thus, the permeability coefficients decrease with the membrane fluidity in the order $L_d > L_o > L_\beta$, each transition leading to an order of magnitude change in the permeation for water and weak acids.

**Differences in permeability of DPPC membranes with ergosterol and cholesterol reflect their phase behavior.** The transitions between $L_o$ and $L_\beta$ phases differ for DPPC membranes with ergosterol and cholesterol. While 15 mol% of cholesterol in DPPC membranes increases the permeability coefficient for formic acid in the experiments approximately 15-fold, the increase is only 5-fold for the same amount of ergosterol (Fig. 4A and C). Also, different from cholesterol, increasing the concentration of ergosterol in DPPC membranes up to 45% leads to a gradual increase of the permeability coefficient for formic acid rather than a more abrupt increase as seen between 0 and 15% cholesterol, suggesting a smooth change from $L_\beta$ to $L_o$ through phase coexistence.

A DPPC membrane with 15 or 30 mol% of cholesterol or ergosterol coexists in $L_\beta$ and $L_o$ phases (Supplementary Table 4) as confirmed by our DSC measurements (Supplementary Fig. 4B and C). The DPPC membrane with 15% of cholesterol forms a stable $L_\beta$ phase in our MD simulations, but the DPPC membrane with 30% cholesterol forms a stable $L_o$ phase, yielding a large change in the free energy profile and a corresponding increase in the permeability coefficient. Comparison of the changes of the permeability coefficients between simulations and experiments suggests that ergosterol has a higher tendency to form a $L_\beta$ phase than cholesterol[37] at the same concentration; cholesterol preferably forms a Lo phase. This is well in line with the observed overall smaller effects of ergosterol on the lipid bilayer structure compared to that of cholesterol[20,32,33]. It may also explain the intriguing behavior of L-lactic acid, which has a much lower (almost 50 fold lower) permeability coefficient in POPC membranes than formic acid, while permeation of L-lactic acid is not detectable in DPPC membranes w/o cholesterol (Fig. 4A and Supplementary Table 4). However, we did detect a measurable permeation of L-lactic acid in DPPC vesicles with 30% and 45%

of ergosterol (Fig. 4C). This observation is in accordance with the known higher condensing effect of cholesterol compared to ergosterol and that cholesterol forms dimeric or even tetrameric aggregates at concentrations above 20%, which may be responsible for the observed permeation slowdown compared to ergosterol[32,33,38,39].

## Discussion

We have used kinetic flux measurements and MD simulations to systematically analyze the permeability of small molecules of different polarity and size through membranes of varying lipid compositions. The used lipid compositions span different head groups, acyl chain lengths, degrees of unsaturation and the effects of cholesterol and ergosterol, allowing us to study the permeation through membranes in various phases, namely $L_d$, $L_o$ and $L_\beta$.

Our MD simulations show that the partitioning of hydrophilic solutes to the membrane interior is comparable to their partitioning in octanol, but hydrophobic compounds partition into the membrane less than they do in octanol (Fig. 1). This highlights the limitations of using the octanol-water partition coefficient ($P_{OW}$) for estimating the permeability coefficients of hydrophobic compounds. The differences in partitioning of hydrophilic and hydrophobic solutes are reflected in the corresponding free energy and solvent proximity profiles (Fig. 1C and D). Unlike bulk solvents which are used to measure the partition coefficient such as $P_{OW}$, membranes are comparably thin anisotropic layers. Hence, the process of solute permeation occurs for a large part through the mixed region at the interface including both the polar solvent and the hydrophobic lipid tails affecting the permeability coefficient (Fig. 1D).

The membrane properties and the phase behavior in particular have a great influence on the permeability coefficients for small polar molecules. We show that in the case of fluid membranes their thickness is the main parameter of the permeability coefficient (Fig. 2 and Supplementary Fig. 7). We find no general correlations between permeability coefficients and lipid surface areas contrary to what was observed by Mathai and colleagues[4]. Instead, we observe that adding carbon atoms to the lipid tails progressively increases the membrane hydrophobic thickness and decreases the permeability, on average, ca. 1.5 fold for every two carbons in the range from 14 to 22 carbons.

We further demonstrate the dependence of the permeability coefficient on the membrane hydrophobic thickness also in membranes with different degrees of unsaturation and sterol content (Supplementary Fig. 6). In contrast, lipid headgroup composition has only limited impact on the permeation of small molecules (Supplementary Fig. 8 and Supplementary Table 5).

The relationship between permeability and membrane physical state ($L_d$ and $L_o$ phases) has received some attention in the past decade[18,40]. In line with these studies, we observe that the fluidity of the membranes decreases (lipid tail order increases) with decreasing degree of lipid tail unsaturation[41,42], and the rate of permeation decreases for both monounsaturated as well as polyunsaturated lipid bilayers[4,14,43]. The effect on solute permeability of changing the lipid unsaturation index ($d$) is relatively small for membranes in the fluid phases $L_d$ and $L_o$, and the difference in permeability coefficient between POPC ($d = 0.5$) and DOPC ($d = 1.0$) membranes is only about 5-fold. Changes of much larger magnitude appear with membrane phase transitions. By comparing liposomes of pure DPPC and pure DHPC at temperatures below and above phase transition, Guler and colleagues[27] showed a difference in water permeability of ca. 2000 fold between the fluid phase ($L_d$) and the gel phase ($L_\beta$). Analogously, we here find a 200 fold difference in water permeability between DOPC ($L_d$) and DPPC liposomes ($L_\beta$) at 20 °C. We also

**Table 1 Permeability coefficients in cm/s of water, formic acids, lactic acid and glycerol for lipid vesicles in different physical states and yeast cells.**

| | $P_{DOPC} \times 10^{-5}$ (cm/s) | $P_{DPPC} \times 10^{-5}$ (cm/s) | $P_{DPPC+15\%cholesterol} \times 10^{-5}$ (cm/s) | $P_{DPPC+15\%ergosterol} \times 10^{-5}$ (cm/s) | $P_{yeast} \times 10^{-5}$ (cm/s) |
|---|---|---|---|---|---|
| Water | 1600 ± 170 | 7.8 ± 1.0 | 23 ± 4 | 12 ± 2 | 10–20[a] |
| Formic acid | 719 ± 172 | 0.27 ± 0.05 | 4.1 ± 0.3 | 1.3 ± 0.1 | 1.1 ± 0.1[b] |
| L-lactic acid | 19.8 ± 3.3 | Not observed | Not observed | Not observed | Not observed |
| Glycerol | 2.3 ± 0.02 | 0.00011 ± 0.00002 | 0.001284 ± 0.00013 | / | – |

[a]Permeability coefficient for water in double mutant *aqy1 aqy2*[76].
[b]Permeability coefficient for formic acid in RA380[14].

observe changes of three orders of magnitude for the permeability of formic acid through membranes in $L_d$ and $L_\beta$ phase, and between one and two orders of magnitude through membranes in $L_o$ and $L_\beta$ phases.

The exact values depend on the membrane composition and on the proximity to the phase transition temperature. We show in our simulations that the large differences in permeability arise from the different free energy profiles of the respective membrane phases, which acquire small shoulders at the membrane-water interface when sterols are present, in line with existing studies[18]. In addition, the pure DPPC membrane ($L_\beta$ phase) shows highly decreased diffusivity through its hydrophobic core (Supplementary Fig. 3). Adding sterols to the DPPC membrane without changing the membrane overall phase leads to an increase in the diffusivity towards the level of membranes in $L_o$. Importantly, the sterols prevent the solutes to drag solvent molecules into the hydrophobic region of the membrane, thereby increasing the hydrophobic thickness (Fig. 4E).

Our measurements reveal important differences between cholesterol and ergosterol, which relate to the differences these sterols have on the phase behavior of the membrane. If different lipid phases coexist or the lipid bilayer is close to its phase transition temperature, the majority of solutes will diffuse in or out through the most permeable parts of membrane and/or the interphase with the coexisting phase[44]. Our results support the view that ergosterol has a higher tendency to form a $L_\beta$ phase[37], while cholesterol forms an $L_o$ phase at the same concentration. This is in line with the observed overall smaller effects of ergosterol on the lipid bilayer structure compared to that of cholesterol[20,32,33].

The plasma membrane of yeast is laterally heterogeneous with a complex organization, specialized compartments and highly ordered rigid domains, which have inspired the here-presented measurements on the permeation of small hydrophilic solutes through membranes in different physical states ($L_d$, $L_o$ and $L_\beta$). We and others have found that the lateral diffusion coefficient of proteins in the plasma membrane of yeast is 3-orders of magnitude slower than that of similar size proteins in the ER or vacuolar membranes[45–48]. The lateral diffusion of proteins in the plasma membrane of yeast is also much slower than in e.g. bacterial membranes[49]. Moreover, we have observed that the permeability of the yeast plasma membrane for formic acid and acetic acid is two to three orders of magnitude lower than that of synthetic lipid vesicles, and that the yeast membrane is virtually impermeable (at the level of passive diffusion) for lactic acid and glycerol, whereas bacterial membranes rapidly permeate these molecules[14].

In our previous work[48], we have hypothesized that this difference in protein diffusivity and solute permeability is due to a more ordered state of the yeast plasma membrane. Specifically, the presence in the PM of long saturated acyl chain(s) sphingolipids, namely phosphoceramide (IPC), mannosyl-inositol-phosphoceramide (MIPC), mannosyl-(inositolphospho)2-ceramide (M(IP)2 C), and ergosterol,

particularly concentrated in the cytoplasmic leaflet[50], could lead to a highly ordered structure. This hypothesis is in accordance with the work of Aresta-Branco and colleagues[19], who studied plasma membrane of *Saccharomyces cerevisiae* by performing anisotropy measurements of diphenylhexatriene and fluorescent lifetime measurements of trans-parinaric acid. Estimates of global membrane order in wildtype cells and studies of mutants defective in sphingolipid or ergosterol synthesis suggest that the yeast plasma membrane harbors highly-ordered domains enriched in sphingolipids. In fact, the observed fluorescence lifetimes (>30 ns) are typical of the gel phase of synthetic membranes[51–54].

In recent work[55], we observed a high degree of unsaturated acyl chains and low values of ergosterol in the very small shell of lipids surrounding membrane transport proteins. These proteins with the so-called periprotein lipidome are embedded in an environment of lipids that are enriched in ergosterol and possibly saturated long-chain fatty acids such as present in IPC, MIPC and M(IP)₂C), which yield a highly liquid-ordered state. The highly ordered state likely explains the slow lateral diffusion and the low solute permeability of the yeast plasma membrane. The ordered state may also form the basis for the robustness of yeast to strive in environments of low pH, high concentrations of ethanol or weak acids, and relate to the ability of *S. cerevisiae* cells to retain glycerol and use it as main osmoprotectant[56–61].

We now show that the permeability coefficients for formic acid, lactic acid, and water in DPPC DPPC/ergosterol and DPPC/cholesterol vesicles are in line with the observations on permeation and lateral diffusion made for the yeast plasma membrane (Table 1). We were not able to test the long saturated acyl chain(s) sphingolipids of yeast, because IPC, MIPC and M(IP)2 C are not available and we used DPPC instead. The permeability of the yeast plasma membrane for water and formic acid is at least two orders of magnitude slower than of vesicles in the liquid disordered phase, i.e., DOPC vesicles. On the other hand, membranes in the gel and liquid-ordered phase, i.e., DPPC vesicles with or without 15% ergosterol or cholesterol, display permeability coefficients similar to that of the yeast plasma membrane. We also notice that the water/formic acid and water/glycerol permeability ratio follow the order $L_d < L_o < L_\beta$' (e.g., DOPC < DPPC + 30% sterol < DPPC). This indicates that formic acid and glycerol permeation through ordered bilayers is more penalized compared to water diffusion. Interestingly, in yeast we observe a water/formic acid ratio that lays between the ones of pure DPPC and DPPC + 30% cholesterol. We have also shown that yeast PM is impermeable to lactic acid in the timespan of 2.5 hours. To our knowledge, passive influx of lactic acid has not been observed in *S. cerevisiae* cells, which parallels the observation that we do not observe lactic acid permeation in vesicles composed of DPPC or DPPC:cholesterol. Yet, we observe slow permeation in vesicles of DPPC:ergosterol (=70:30, 55:45), where the Lo phase predominates. The evidence collectively corroborates that the yeast plasma membrane behaves as a highly ordered barrier to the

permeation of small polar molecules, which is different from the more fluid plasma membranes of mammalian and prokaryotic cells. Our data are in line with the suggestion that in yeast the liquid-disordered phase is confined to the periprotein lipids[55]. The high degree of unsaturated acyl chains and low amounts of ergosterol in the periprotein lipidomes may allow sufficient conformational flexibility of the proteins, yet without compromising the exceptional permeability barrier of the membrane. The high degree of saturated lipid in the yeast plasma membrane[62] compared to for instance the membranes of human cells may be seen as a possible evolutionary adaptation of yeasts to ergosterol as their main sterol. It is clear from the differences between ergosterol and cholesterol, that the two sterols exhibit different phase behavior and have different interactions with saturated and unsaturated lipid tails[20,32,33,37]. In agreement, our measurements with saturated DPPC lipids show that cholesterol exhibits large jumps in permeability, fluidity and, hence, in phase behavior, whereas ergosterol impacts the membrane properties and phase state more smoothly. This allows ergosterol to function as a component that can steadily regulate fluidity in the highly rigid yeast plasma membranes but less so in more unsaturated fluid membranes, where cholesterol is a better regulator.

In summary, we find that the membrane thickness and the degree of lipid tail unsaturation have a significant impact on the solute permeability in membranes in the fluid phase, but the biggest changes in permeation happen when these factors lead to the transition from the fluid ($L_d$) to the gel-like phase ($L_\beta$). We observe a drop of three orders of magnitude in the permeability of formic acid and glycerol, and two orders of magnitude in water permeability for DPPC, compared to DOPC membranes. This is mainly due to the large differences in the solubility of the permeating solutes in the membrane interior. The addition of cholesterol or ergosterol to DPPC, which coincides with the formation of the liquid-ordered phase ($L_o$), induces a partial restoration of permeability towards the level of fluid membranes. Our measurements reveal that ergosterol has a small impact on lipid bilayer structure compared to cholesterol, with the latter having a higher tendency to induce a $L_o$ phase at the same concentrations. Finally, we compare our results with in vivo data from *Saccharomyces cerevisiae* presented previously and use the permeability data as reporter of the physical state of the yeast plasma membrane. Our results reveal that the yeast plasma membrane is in a highly rigid physical state comparable to model membranes of DPPC with 0–15% ergosterol at a gel-like $L_\beta$ phase. Moreover, unlike cholesterol, ergosterol changes the membrane properties including the permeability coefficient smoothly with its concentration in that regime allowing it to act as a membrane rigidity regulator in yeasts.

## Methods

**Materials**. The weak acid solutions were prepared using the following salts: sodium-acetate (BioUltra, ≥99.0%; Sigma-Aldrich); sodium-benzoate (BioXtra, ≥99.5%, B3420-250G; Sigma-Aldrich, St. Louis, MO); sodium-butyrate (≥98.5%; Sigma-Aldrich); sodium-formate (BioUltra, ≥99.0%; Sigma-Aldrich); sodium L-lactate (>99.0%; Sigma-Aldrich); sodium-propionate (≥99.0%; Sigma-Aldrich); pyruvic acid-sodium salt (99+%; Acros Organics, Geel, Belgium); potassium-sorbate (purum p.a., ≥99.0%; Sigma-Aldrich): and potassium chloride (pro analyses; BOOM Laboratorium Leveranciers, Meppel, The Netherlands). Glycerol was purchased from BOOM Laboratorium Leveranciers (Meppel, The Netherlands). The following lipids were used and purchased from Avanti Polar Lipids (Alabaster, AL): 1,2-dioleoyl-*sn*-glycero-3-phosphocholine (DOPC); 2-dioleoyl-*sn*-glycero-3-phosphoethanolamine (DOPE); 1,2-dioleoyl-*sn*-glycero-3-phospho-(1′-rac-glycerol) sodium salt (DOPG); 1-palmitoyl-2-oleoyl-*sn*-glycero-3-phosphocholine (POPC); 1-palmitoyl-2-oleoyl-*sn*-glycero-3-phosphoethanolamine (POPE); 1-palmitoyl-2-oleoyl-*sn*-glycero-3-phospho-(1′-rac-glycerol) sodium-salt (POPG); 1,2-dipalmitoyl-*sn*-glycero-3-phosphocholine (DPPC); 1,2-dimyristoleoyl-*sn*-glycero-3-phosphocholine (14:1 Cis) PC), 1,2-dipalmitoleoyl-*sn*-glycero-3-phosphocholine (16:1 Δ9-Cis) PC), 1,2-dieicosenoyl-*sn*-glycero-3-phosphocholine (20:1 Cis)

PC); 1,2-dierucoyl-*sn*-glycero-3-phosphocholine (22:1 (Cis) PC); cholesterol; ergosterol.

**Weak acid solutions**. The 1 M stock solutions (0.5 M for benzoic acid) were prepared by dissolving the salt, or glycerol, into 100 mM potassium phosphate (KPi) and the pH was adjusted to 7.0 using 4 M NaOH. An empirical linear relation (y = mx + q) between osmolyte concentration and osmolality was determined for each solution (Supplementary Fig. 9). The osmolality was measured using a freezing point depression osmometer (Osmomat 3000 basic; Genotec, Berlin, Germany). The empirical relations were used to estimate the osmolyte concentrations needed for an osmolality of ~300 mOsmol/kg, that is, upon mixing with the liposome solution. The stock solutions were accordingly diluted to the desired concentration before the experiment.

**Vesicle preparation**. The lipids were purchased from Avanti Polar Lipids in powder and suspended in chloroform to a concentration of 25 mg/mL. After mixing the solubilized lipids in the desired ratio, a rotary vaporizer (rotavapor r-3 BUCHI, Flawil, Switzerland) was used to remove chloroform by evaporation. Next, the lipids were suspended in diethylether and subjected to a second of evaporation. Finally, the lipids were hydrated in the assay buffer (100 mM KPi, pH 7) and adjusted to a concentration of 10 mg/mL. The lipid solution was homogenized by tip (3.18 mm) sonication with a Sonics Vibra Cell sonicator (Sonics & Materials Inc. Newtown, CT, USA) at 4 °C (ice water) for 4 min with 15 s pulses and 15 s pause between every pulse. Amplitude of the sonicator was set to 100%. The prepared vesicles were stocked at 20 mg/mL in liquid nitrogen to prevent oxidation.

**Preparation of vesicles filled with calcein**. The fluorophore calcein (from Sigma-Aldrich) was solubilized at a concentration of 100 mM with 50 mM KPi, and the pH was adjusted to 7.0 using aliquots of 4 M KOH. The stocked vesicles (2 mg of lipid) were pelleted by ultracentrifugation (280,000 × *g*, 4 °C, 20 min with a TLA 100.1 rotor in a Beckman Optima TLX Ultracentrifuge; Beckman Coulter Life Sciences, Indianapolis, IN) and resuspended in 0.9 mL of 89 mM KPi, pH 7.0. Calcein was added to the liposome solution at a self-quenching concentration (10 mM) and enclosed in the vesicles by 3 cycles of rapid freezing in liquid nitrogen and thawing at 40 °C (or 60 °C for mixtures containing DPPC). Thus, the osmolality of the liposomutle lumen (filled with 10 mM calcein plus 89 mM KPi pH 7.0) is ~190 mOsmol/kg, which equals the osmolality of the assay buffer (100 mM KPi pH 7.0). After extrusion through a 200 nm polycarbonate filter at 20 °C (or 60 °C for mixtures containing DPPC) to homogenize the vesicles, they were eluted through a 22-cm-long Sephadex-G75 (Sigma-Aldrich) column pre-equilibrated with the assay buffer to remove the external calcein. The collected 1 mL fractions containing the calcein-filled vesicles were identified by eye using an ultraviolet lamp (for fluorophore excitation) and diluted in a total volume of 10 mL of the assay buffer.

**Stopped-flow experiments**. A stopped-flow apparatus (SX20; Applied Photophysics, Leatherhead, Surrey, UK) operated in single-mixing mode was used to measure fluorescence intensity kinetics upon application of an osmotic shock to the vesicles filled with calcein. To impose the osmotic shock, the solution of the permeant, weak acid in most cases (ca. 100 mM of sodium or potassium salt of the weak acid in 100 mM KPi pH 7.0; ~300 mOsmol/kg after mixing), and the vesicles were loaded each in one syringe and forced first through the mixer (1:1 mixing ratio with 2 ms dead time) into the optical cell (20 μL volume and 2 mm pathlength). The temperature of the optical cell was set at 20 °C using a water bath. The white light emitted by a xenon arc lamp (150 W) was passed through a high-precision monochromator and directed to the optical cell via an optical fiber. The band pass of the monochromator was optimized and set to 0.5 nm (for calcein) to prevent fluorophore photobleaching during the experiment. Calcein was excited at 495 nm. The emitted light, collected at 90°, was filtered by a Schott long-pass filter (cutoff wavelength at 515 nm) and detected by a photomultiplier tube (R6095; Hamamatsu, Hamamatsu City, Japan) with 10 μs time resolution. The voltage of the photomultiplier was automatically selected and kept constant during each set of experiments. The fluorescence intensity kinetics after the osmotic shock was recorded with logarithmically spaced time points to better resolve faster processes. For noise reduction, multiple acquisitions (three for slow kinetics and nine for fast kinetics) were performed for each experimental condition. The stopped-flow apparatus was operated with the built-in software "Pro Data SX", and the fluorescence traces visualized via the built-in software "Pro-Data viewer". After acquisition, the generated files were converted to ASCii format using "APL Pro-Data Converter".

**Preprocessing of the in vitro kinetic data**. The raw data were preprocessed in MATLAB (R2018a; The MathWorks, Natick, MA) for further analysis. First, the N number of curves, which we called fi(t), acquired with a single experimental condition, were averaged ($F(t) = N - 1\sum fi(t)$) to reduce the noise. For calcein, the resulting kinetic curves $F(t)$ were normalized to 1 at time zero ($F(t)/F(0)$), i.e., the mixer dead time ($t_0 = 2$ ms).

**Size distribution of vesicles**. The size distribution of vesicles was measured by Dynamic Light Scattering (DLS) using the DynaPro NanoStar Detector (Wyatt Technology, Santa Barbara, CA) operated via the built-in software DYNAMICS. Empty vesicles were prepared starting from 1 mg of lipids by three freeze-and-thaw cycles at 40 °C (or 60 °C for mixtures containing DPPC). After 13 times extrusion through a 200 nm filter, vesicles were eluted through a 22-cm-long Sephadex-G75 column pre-equilibrated with 100 mM KPi (pH 7.0). Before the DLS measurements, the vesicles were diluted with the assay buffer to a concentration in the range from 2 µg/mL to 2 mg/mL. Measurements were performed with a scattering angle of 90°. For each measurement, at least 10 acquisitions of 20 s each were performed at a temperature of 20 °C. For each acquisition, at least 2 million counts were recorded. The correlation curves and the intensity-weighted distributions were obtained with the built-in analysis software.

**Fit of the in vitro kinetics**. We assume that (i) the surface area of the vesicles is fixed and freely deformable, (ii) the membrane thickness is much smaller than the vesicle radius, (iii) all vesicles share the same exact membrane composition, (iv) membrane composition is nanoscopically homogeneous, (v) calcein is homogeneously distributed inside the vesicles, (vi) osmolyte solutions are well mixed, and electrically neutral, and (vii) the external solutions are an infinite source of molecules. A detailed description of the model is presented in work[15].

We calculate the normalized ratio $<F(t)>/<F(0)>$, that is the time evolution of calcein fluorescence, as in Appendix B of work[15]. Briefly, the relaxation kinetics of the calcein concentration $c(r_{0,t})$ was computed by numerical solution of the system of differential equations describing the dynamics of a spherical vesicle of radius $r_0$ upon osmotic upshift. The numerical solution was used to calculate the ratio $F(r_{0,t})/F(0)$, using the Stern-Volmer equation accordingly modified:

$$\frac{F(t)}{F(0)} = \frac{1 + K_{SV}\, c(0)}{1 + K_{SV}\, c(t)}, \tag{1}$$

where $K_{SV}$ (M$^{-1}$) is dynamic quenching constant of calcein. To calculate the permeability coefficients of water ($P_W$) and osmolytes ($P_O$), the time evolution of calcein fluorescence, $<F(t)>/<F(0)>$ was fitted to the kinetic curves. The population-averaged ratio $<F(t)>/<F(0)>$ was computed by using the vesicle size distribution measured in dynamic light scattering (DLS, Supplementary Fig. 10) experiments and fitted to the experimental data using the FMINUIT[63] minimization routine. For the "impermeable" osmolyte (KCl), two fitting parameters were used: the quenching constant $K_{SV}$ and the water permeability coefficient $P_w$ (cm/s). For the permeable osmolytes (Table 2), $K_{SV}$, $P_W$, and the osmolyte permeability coefficient $P_O$ were fitted to each other. Exemplary fits for water, formic acid, lactic acid and glycerol in liposomes of pure DOPC, POPC, DPPC, and DPPC + 45% ergosterol are shown in Supplementary Fig. 11 and the relative fitting parameters are shown in Supplementary Table 6. To improve the accuracy and to estimate the error of P, we repeated the fit for each of the 10 size distributions acquired by DLS. The mean of the fitted values was used as the best estimate of P, and the standard deviation indicates the experimental uncertainty for the permeability coefficient δP, which derives mostly from the ambiguity of the vesicle size distribution estimation by DLS. The other parameters required for calculation of $c_2(r_{0,t})$ were set to their experimental values, which are $pH_{out} = 7$, $[KPi]_{in} = 89$ mM, $[KPi]_{out} = 100$ mM, $c_2(r_{0,0}) = 10$ mM, $M_W$ H$_2$O = 18 cm$^3$/mol, $pK_a$ (KPi) = 7.21, and see Table 2 for $pK_a$ (acid). $M_W$ H$_2$O is the molar volume of water. The concentration of the osmolyte $[O]_{out}$ in the external solution was set to 40–50-mM for all compounds, except for glycerol, which was set to 120 mM, as obtained from Supplementary Fig. 9. For each of the tested weak acids, the total

### Table 2 Molecular Weight and $pK_a$ values of the used osmolytes.

| Osmolyte | $M_W$ (g/mol) | $pK_a$ (25 °C) | Compound ID |
|---|---|---|---|
| KCl | 74.55 | N/A | |
| Sodium acetate | 82.03 | 4.76 | 176 |
| Sodium benzoate | 144.1 | 4.19 | 243 |
| Sodium butyrate | 110.09 | 4.82 | 264 |
| Sodium formate | 68.01 | 3.75 | 284 |
| Sodium L-lactate | 112.06 | 3.86 | 612 |
| Sodium propionate | 96.06 | 4.88 | 1032 |
| Sodium pyruvate | 110 | 2.45 | 1060 |
| Potassium sorbate | 150.22 | 4.76 | 643460 |
| Glycerol | 92.09 | 14.4 | 753 |

N/A not applicable.
The $pK_a$ values were taken from the PubChem database (https://pubchem.ncbi.nlm.nih.gov/), using the compound ID indicated in the last column.

osmolyte concentration is $2[O]_{out}$ to account for the counterion released by the weak acid salt.

**Differential scanning calorimetry (DSC)**. Differential scanning calorimetry (DSC) measurements were performed using an MC-2 calorimeter (MicroCal, Amherst, MA) to perform ascending and descending temperature mode operations. The lipid concentration used was 1–2 mg/mL and the temperature of the sample and reference cells was controlled by a circulating water bath. The scan rate was 20 °C/h for both heating and cooling scans. Data were analyzed using ORIGIN software provided by MicroCal. Samples were scanned 5 times to ensure the reproducibility of the endotherms.

**Molecular dynamics (MD) simulations**. MD simulation is an established method to study permeability of solutes through lipid membranes[9,16–18]. Here we rely on the latest version of the CG Martini model[25]. Bereau and co-workers have shown that the Martini model is very accurate for computing permeation rates, with good correlations to both all-atom simulations and experimental measurements over a wide range of compounds[23,24]. In the current work, permeants are modeled as generic single-particle solvents of varying hydrophobicity from I (most hydrophilic) to IX (most hydrophobic). The initial conditions of the simulated lipid bilayers were generated using the tools Insane and Martinate[64] to yield lateral dimensions of $10 \times 10$ nm with the bilayer repeat distance of 10 nm. The systems were equilibrated at their respective temperatures for at least 10 ns prior to production simulations. A standard simulation setup for the Martini model was used as described in previous work[25]. In brief, the simulation temperature was coupled to a v-rescale thermostat[65] at room temperature of 293 K, separately for the lipids and solvent. A Parinello-Rahman barostat[66] was used for pressure coupling at 1 bar with a coupling constant of 24 ps independently for the membrane plane and its normal. Standard time step of 20 fs was used for all simulations and the trajectory was recorded every 1 ns. For production, we have simulated all systems for 5 µs to ensure convergence; simulations of membranes at the gel phase were run for 10 µs. Simulations were run using GROMACS simulation package ver. 2019.3 in a mixed precision compilation without GPU support[66,67].

Adaptive weighted histogram (AWH) method[68] was used to obtain free energy profiles of the permeating particles. The AWH weight factors were updated every 0.1 ns. Particle permeation was done along the z-axis as a membrane normal. Distance of the permeating particle from the membrane center was calculated using the cylindrical coordinate, which uses the local membrane lipids within a cutoff distance of 1.0 nm to calculate the center of mass of the membrane[67]. The force constant of the biasing potentials in AWH was 2000 kJ mol$^{-1}$ nm$^{-2}$ and the coupling constant between the generalized and the molecular coordinates was 10,000 kJ mol$^{-1}$ nm$^{-2}$. The constant of local friction was calculated from force autocorrelation implemented within the AWH method[67,68]. The friction profiles were denoised using a Butterworth lowpass filter[69] of the fourth order with a Nyquist frequency of 100 Hz. Error estimate was calculated from the average noise around the denoised curve and from the deviations from the symmetrical shape between the left and right part of the profile. The profiles of solvent proximity to the permeants were calculated using GROMACS tool gmx mindist with a cutoff radius of 0.6 nm between the permeant and solvent molecules. Scripts used to generate and analyze the simulations are available in an open public repository[70].

**Inhomogeneous solubility-diffusion model**. The inhomogeneous solubility-diffusion (ISD) model was used to calculate the permeability coefficient $P$ using Eq. 2

$$\frac{1}{P} = \int_{-h/2}^{h/2} \exp\left(\frac{\Delta F(z)}{k_B T}\right) \frac{f(z)}{k_B T} dz, \tag{2}$$

where $F(z)$ denotes the free energy as a function of the membrane normal, $f(z)$ denotes the local friction, $h$ is the thickness of the membrane, and $\beta = 1/(k_B T)$ with the Boltzmann constant $k_B$ and temperature $T$. Local friction, $f(z)$, is inversely proportional to the local diffusivity as $f(z) = k_B T/D(z)$. Error estimates were calculated using standard chain rules for propagating errors, which arise mainly from the uncertainty of the free energy profiles.

The permeation dynamics is assumed to be diffusive in the ISD model. Deviations from such an assumption may lead to inaccurate estimates of the permeability coefficients for permeants that interact strongly with one another or with the membrane[71,72]. Also, for higher concentrations, the ISD model does not capture the collective behavior of the permeating solute and its effects on the membrane. However, for compounds at low concentrations the estimates of the permeability coefficients from the ISD model are comparable to the unbiased estimates from transition-based counting[72]. Further studies on the permeability of small solutes through lipid membranes and the ISD in various scenarios are provided in the references[16,71–74].

By assuming average behavior over the permeation pathway in Eq. 2, we obtain a simpler homogeneous solubility diffusion (HSD) model, also known as the Meyer-Overton rule

$$P = \frac{KD}{h}, \tag{3}$$

with $K$ denoting the solubility constant, which is equal to the partitioning coefficient of the permeant between water and the membrane ($P_{MW}$). Other symbols have the same meaning as above in Eq. 2.

**Additional analysis of MD data.** Solvent accessibility profiles represent the local probability of the permeating molecule to be in a contact (within cutoff distance of 0.6 nm) with at least one solvent molecule at the given distance from the membrane center. The water density profile $\rho(z)$ and the solvent accessibility profile are related to one another as they both describe the spatial distribution of water molecules around the membrane. The definitions of the profiles are, however, different. Water density describes the probability density of water with respect to the distance from the membrane center. Solvent accessibility follows the permeating solvent and shows the probability of at least one water molecule to be present within the solvation shell of the permeant. This also includes the capacity of the permeant to drag/repel water molecules along its permeation path through the membrane—an effect that is not included in the water density profile $\rho(z)$.

Membrane hydrophobic thickness, $h$, was calculated after the definition of Luzzati[75] according to the formula

$$h = L_z - \int_0^{L_z} \rho'(z)\,dz, \qquad (4)$$

where $L_z$ is the simulation box length in the $z$-direction (also known as bilayer repeat distance), and $\rho'(z)$ is the local water density normalized to the value in the bulk.

The partition coefficients were obtained directly from the free energy profiles as the difference

$$\Delta F_{MW} = F(\text{membrane}) - F(\text{solvent}), \qquad (5)$$

where $F(z)$ is the calculated free energy at the membrane center and in the solvent bulk phase, respectively. The octanol-water partition coefficients were taken directly from ref. [25].

**Reporting summary**. Further information on research design is available in the Nature Research Reporting Summary linked to this article.

## Data availability

All generated data is available in the main text and/or in the associated Supplementary information file. Minimal source data and materials used in the analysis are available in public repositories https://doi.org/10.5281/zenodo.6010416 for MD simulations, and https://github.com/jacopofrallicciardi/Membrane-permeability-Source-data for experimental measurements.

## Code availability

Scripts used to analyze experimental data and calculate permeability coefficients are available in an open public repository (doi.org/10.5281/zenodo.5176264). Scripts used to generate and analyze the simulations are available in an open public repository (doi.org/10.5281/zenodo.5032419).

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

## Acknowledgements

We thank Matteo Gabba for insightful discussions at the start of the project. The research was funded by an ERC Advanced grant (ABCVolume; #670578) and the EU CoFund program ALERT to B.P., and a ERACoBioTech grant (MeMBrane) from NWO to S.J.M. We thank SURFsara and the Center for Information Technology of the University of Groningen for their support and for providing access to the high-performance computing clusters Cartesius and Peregrine.

## Author contributions

J.F., J.M., S.J.M., and B.P. designed the studies and wrote the manuscript. J.F. and P.S. performed the stopped-flow measurements and analyzed the relative data. J.M. performed the molecular dynamics simulations and analyzed the relative data. B.P. and S.J.M. supervised the work.

## Competing interests

The authors declare no competing interests.

## Additional information

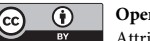

