## [Peer Review File · Nature Communications]

Membrane thickness, lipid phase and sterol type are determining factors in the permeability of membranes to small solutesREVIEWER COMMENTS

Reviewer #1 (Remarks to the Author):

General

The paper investigates the permeability of several permeants through DPPC, POPC, or mixed POPC-DPPC membranes with a varying concentration of sterols (cholesterol or ergosterol). The large change in permeability with increasing sterol concentration in DPPC is linked to a phase transition. Next, this observation is used to interpret the permeability of yeast plasma membrane as a sensor for this plasma membrane being in a gel-like phase.

This paper is a very timely contribution, because it presents the permeability as an indirect reporter of the lipid phase. In the future, it could be that other permeants may be proposed as more adequate probes for the membrane phase, so the scientific value is that it may be the starting point for follow up research, since there is still a large ongoing debate regarding the biological importance of membrane permeability. The scientific methods seem sound and appropriate. The combination of both experimental work and computational work is a strong point in this paper.

A major point of criticism is that the paper presents that the finding about permeability decreasing following " $L_d > L_o > L_{\beta}$ " (page 11) as a new finding in its abstract/introduction/text/conclusion. However, it was already known that " $L_d \gg L_{\beta}$ " for water permeability through DPPC (experimental, see Guler, S. D. et al. Effects of ether vs. ester linkage on lipid bilayer structure and water permeability. *Chem. Phys. Lipids* 160, 33–44 (2009)) and also " $L_d > L_o > L_{\beta}$ " for water and oxygen permeability (computational, see Ref. 18 that is indeed cited). The authors need to rephrase their abstract/introduction/text/conclusion to acknowledge these earlier findings properly and put their work in context. (It is of course true that the present work is more elaborate than these earlier studies.)

The presentation needs to be improved, for adding clarity, solving inaccuracies e.g. regarding whether data is experimental or from simulations, improving reproducibility. Hence a major revision is asked for, which will however not change the essential content of the manuscript.

Comments

- * Temperature of experiment/simulations should be mentioned already in the main text (maybe abstract/conclusion) since it is crucial for the present study that it is below the melting temperature of DPPC.
 - * Page 3, When mentioning the solubility diffusion model, does this concern the inhomogeneous or homogeneous model? Please specify in the text.
 - * Page 4, $\log(Pow)$ is not defined in its first mention. When it is defined in the next sentence, I suspect that "logarithm of" is missing.
 - * I am curious whether the solvent accessibility (Fig. 1, definition in Methods section) is different from $\rho'(z)$ (Methods section)? I would expect that the $\rho'(z)$ curve contains the same information as the solvent accessibility?
 - * How are Pow and Pmw obtained in Figure 1E and 1F? I assume it is experimental in 1E and simulations in 1F? Please add this info to the caption.
- If I understand correctly, the Pow and Pmw values in Table S.I were taken from the Martini-3 publication. On page 23, the notation K is used for the partitioning coefficient,

which refers to the (computational) free energy difference between the membrane center and water bulk phase. Is this the same quantity? Please clarify in the text. I did not see explicitly in the text how the P_{ow} values in Table S.I and in Fig. 1E were measured in experiment. Please clarify this in the manuscript. In short, I believe that differences between $P_{ow} / P_{mw} / K$, experimental/computational, can have quite some influence on their values, especially since membranes are inhomogeneous and anisotropic, which is more (or less) ignored depending on the quantity. This is why I would like to see this presented more clearly.

* In general, the authors should to go through the manuscript and be more rigorous about mentioning whether presented data in text/figures/supp. info., and the conclusions thereof, are experimental or simulated. This is easy to do for the authors, and it will help to cite the research in this paper in a correcter way.

* I don't find in the text how the presented permeability data is calculated from simulations. Was it equation 1 or equation 2?

* In several figures, the data are presented without mentioning the permeant. For instance, Fig. 1F, Fig. 2C, etc. Same for the supporting info.

* Page 6, top. For hydrophobic permeants, the deviation of the trend of P versus P_{mw} can also be due to barriers in the membrane head group region. A free energy barrier of 1 kBT (not discussed in the text) changes the permeability (P) more than a free energy well of 1 kBT (reflected by $\log P_{mw}$). See e.g. <https://doi.org/10.1021/acs.jctc.7b00039> The authors can mention the influence of free energy barriers for hydrophobic permeants in their text as a possible cause.

* Page 6. "Increasing the length ... increases the hydrophobic thickness of the membrane (Fig. 2B)." This is not visible to me in Fig. 2B?

* Page 9, top. I thought that there would already be a tertiary phase diagram known for the DPPC, DOPC, cholesterol mix (e.g. ref 26). How does the finding compare with previously published phase diagrams? Is it quantitatively the same, or only qualitatively? I also suggest adding some indication on Fig. 4A/B/C where is the coexistence region, e.g. to see the phase diagram 'superimposed' on these graphs, e.g. with some arrows. That would really tie

together the message of the phase transition being responsible for the transition. (This is an optional suggestion.)

* I am wondering why the choice made to represent permeability ratio's in Fig. 2A/B, 3A/B, etc.? What was there to gain compared to plotting the real values? Maybe the reason can be added to the text.

* Page 21, Fit of the in vitro kinetics: this subsection is dense. Adding information will help to make it reproducible. Please add the Stern-Vomer equation or the specific equation that is fit, such that the reader can relate the equation to the 'fit' parameters and 'fixed' parameters mentioned in this section. Please add examples of the in vitro kinetics (maybe together with their fit) in the Supp. Info. Please add a few notes on the assumptions that are in this fit (e.g. homogeneous distribution inside liposome? membrane thickness known or neglected? etc.) or refer to a paper that lists all the assumptions and limitations. (This last point is valuable to put the results in perspective, because experimental permeabilities can be quite error prone. Computational permeabilities as well, by the way.)

* Formula's are not rendered correctly in the manuscript, unfortunately, especially pages 20-24. I've not been able to review formula's.

* Page 11, "... these cavities are filled with ergosterol or cholesterol when sterols are present in the membrane" Can this statement be backed up? E.g. with a radial distribution function between phospholipids and sterols, or with a free volume graph? Otherwise, can it be made clear

that this is speculation?

* Page 17, "when more phases coexist, solutes preferentially permeate through the more fluid parts of the membrane an/or through the region between the coexisting phases." While this claim is very plausible, it is not written in the paper on what observation this claim is based. How was this assessed?

* I do not find Eq. 3 back in the cited reference 71 by Luzzati. Can you point me to the correct location in that paper or to the correct citation?

* Page 12, "while its permeation is not detectable" Whose permeation? L-lactic acid or formic acid?

* Page 13, "these differences are reflected in". Which differences are meant here? Please rephrase slightly.

* Page 13, "we observe that the fluidity of the membranes decreases" Please add a definition and/or explanation what is meant explicitly with 'fluidity'. Also, was this observation done in the current work or in the work that is cited at the end of the sentence? In the latter case, it would be better to write something like "we observed in other work..." If it was observed in this work, the "fluidity" results should be added to the manuscript or Supp. Info.

* Page 14, "We have hypothesized that this difference in protein diffusivity..." This seems to refer to earlier work, please add the citation. If it is a hypothesis in this specific work, the sentence needs to be rephrased somewhat.

* Page 15, "We now show that..." The citation to ref 56 is strange. Aren't these permeabilities the experimental results of the present paper?

* Fig. 3C, 4D, 4E have very poor resolution.

* I thought Lo had a non-capital "o" as subscript.

* Citation 58 is empty

Reviewer #2 (Remarks to the Author):

Frallicciardi and coauthors have studied the permeation of water and of different solutes (weak acids and glycerol) through synthetic membranes, as function of lipid chain length, chain unsaturation and also membrane organization in the presence of cholesterol and ergosterol. One of their motivation is to understand the difference in the permeability of the membrane of yeasts and of mammalian cells. The originality of their work is the combination of experiments - a fluorescence-based assay that monitors the volume changes upon osmotic shock on liposomes using a stopped-flow system and coarse-grained MD simulations (already developed in their BJ paper (ref. 20))- with the last version (3) of the Martini model. With these simulations, it is possible to have details about the free energy profiles across the membrane and the localization of the water molecules in the presence of the solutes.

They have found a relatively good agreement between the experiments and the simulations, which confirms that the Martini force-field correctly models the transfer of

these polar components through membranes. They have also shown that the permeation of these solutes depends on the membrane thickness and on lipid tail unsaturation when the membrane is in a Ld (liquid disordered) state. But, they show that the state of the membrane, Ld, versus Lo (liquid ordered) versus gel has a much stronger effect, which is not really surprising. They also confirm that the phase diagrams with ergosterol and cholesterol are different with the same lipids and thus the solute permeation. Globally, the work is well-performed and the results certainly useful, but not really surprising. I miss the important message that would justify publication in Nature Communications. In addition, the presentation of the results gives the impression that the authors "discover" that the membrane organization, and thus, the addition of sterols, has an impact on permeation. They describe their results first as an effect of sterol or cholesterol, and only next, consider that strong changes in permeation occur because the addition of these sterols lead to different membrane organizations, when this is an obvious reason. They even perform DSC measurements to measure the transition temperature, when the phase diagrams and these properties are probably available in the literature. It would make more sense to compare the permeation of the solutes for the same phase, in the presence of sterol or cholesterol. The authors also completely ignore the seminal work from E. Evans on water permeability as a function of membrane composition and state (W. Rawicz et al Biophys. J. 94, 4725 (2008); K. Olbrich, et al Biophys. J. 79, 321 (2000)). Altogether, I think that this paper should be published in a more specialized journal, such as Biophysical journal.

Minor

- In general, the temperature of the experiments should be indicated
- Figure 1: Why not also plotting the results of the experiments versus those of the simulations for the same solutes to stress the agreement?
- Page 6: the exact nature of the lipids should be detailed in the text (instead of "lipids with mono-unsaturated tails of lengths between 14 and 26 carbons"), instead of in the Supplementary Information
- Page 6 " Increasing the length of the phospholipid(Fig. 2B)." It should be Fig. 2C
- Figure 2, Legend: the level of unsaturation of the chain and the solute considered for the simulations should be detailed
- Page 7: The definition of the degree of unsaturation is not provided and this term is ambiguous. It would be much better to explain that mixtures of saturated and unsaturated lipids have been studied with different respective fractions, and indicate which lipids
- Fig. S4: the colors do not correspond to the legend

REVIEWER COMMENTS

Reviewer #1 (Remarks to the Author):

General

The paper investigates the permeability of several permeants through DPPC, POPC, or mixed POPC-DPPC membranes with a varying concentration of sterols (cholesterol or ergosterol). The large change in permeability with increasing sterol concentration in DPPC is linked to a phase transition. Next, this observation is used to interpret the permeability of yeast plasma membrane as a sensor for this plasma membrane being in a gel-like phase.

This paper is a very timely contribution, because it presents the permeability as an indirect reporter of the lipid phase. In the future, it could be that other permeants may be proposed as more adequate probes for the membrane phase, so the scientific value is that it may be the starting point for follow up research, since there is still a large ongoing debate regarding the biological importance of membrane permeability. The scientific methods seem sound and appropriate. The combination of both experimental work and computational work is a strong point in this paper.

A major point of criticism is that the paper presents that the finding about permeability decreasing following “Ld > Lo > Lbeta” (page 11) as a new finding in its abstract/introduction/text/conclusion. However, it was already known that “Ld >> Lbeta” for water permeability through DPPC (experimental, see Guler, S. D. et al. Effects of ether vs. ester linkage on lipid bilayer structure and water permeability. Chem. Phys. Lipids 160, 33–44 (2009)) and also “Ld > Lo > Lbeta” for water and oxygen permeability (computational, see Ref. 18 that is indeed cited). The authors need to rephrase their abstract/introduction/text/conclusion to acknowledge these earlier findings properly and put their work in context. (It is of course true that the present work is more elaborate than these earlier studies.)

Reply: We thank the reviewer for her/his thoughtful comments. We are aware of the studies that Reviewer 1 is referring to and apologize for not being sufficiently careful in putting our work in context. We emphasize that our studies are based on weak acid, glycerol and water permeation rather than water only. We have rewritten the corresponding sections of the manuscript; see pages 2, 9, 13 and 14.

The presentation needs to be improved, for adding clarity, solving inaccuracies e.g. regarding whether data is experimental or from simulations, improving reproducibility. Hence a major revision is asked for, which will however not change the essential content of the manuscript.

Reply: We have carefully checked and edited the text for ambiguities in the experimental and simulation data.

Comments

* Temperature of experiment/simulations should be mentioned already in the main text (maybe abstract/conclusion) since it is crucial for the present study that it is below the melting temperature of DPPC.

Reply: We now mention the temperature of the experiments in the introduction of the main text. We also indicate the temperature in the figure and table legends.

* Page 3, When mentioning the solubility diffusion model, does this concern the inhomogeneous or homogeneous model? Please specify in the text.

Reply: The inhomogeneous solubility-diffusion model was used to calculate the permeability coefficients from simulations. This is stated in the Methods section and in the main manuscript text, first sentence in the second paragraph of page 4.

* Page 4, $\log(\text{Pow})$ is not defined in its first mention. When it is defined in the next sentence, I suspect that "logarithm of" is missing.

Reply: Thank you. The log was indeed missing in a few cases. LogP is \log_{10} (partition coefficient), where P refers to the equilibrium distribution of a molecule between the hydrophobic and hydrophilic phase of two immiscible solvents. This definition has been added to the legend of figure 1.

* I am curious whether the solvent accessibility (Fig. 1, definition in Methods section) is different from $\rho'(z)$ (Methods section)? I would expect that the $\rho'(z)$ curve contains the same information as the solvent accessibility?

Reply: The water density profile $\rho(z)$ and the solvent accessibility profile are related to one another as they both describe the spatial distribution of water molecules around the membrane. The definitions of the profiles are, however, different. Water density describes the probability density of water with respect to the distance from the membrane center. Solvent accessibility follows the permeating solvent and shows the probability of at least one water molecule to be present within the solvation shell of the permeant. This also includes the capacity of the permeant to drag/repel water molecules along its permeation path through the membrane—an effect that is not included in the water density profile $\rho(z)$. The above explanation has been added to the Method section on page 26.

* How are Pow and Pmw obtained in Figure 1E and 1F? I assume it is experimental in 1E and simulations in 1F? Please add this info to the caption.

If I understand correctly, the Pow and Pmw values in Table S.I were taken from the Martini-3 publication.

On page 23, the notation K is used for the partitioning coefficient, which refers to the (computational) free energy difference between the membrane center. and water bulk phase. Is this the same quantity? Please clarify in the text. I did not see explicitly in the text how the Pow values in Table S.I and in Fig. 1E were measured in

experiment. Please clarify this in the manuscript. In short, I believe that differences between $\text{Pow} / \text{Pmw} / K$, experimental/computational, can have quite some influence on their values, especially since membranes are inhomogeneous and anisotropic, which is more (or less) ignored depending on the quantity. This is why I would like to see this presented more clearly.

Reply: We now include in the legend of Figure 1 how $\log P$ values were obtained. Briefly, experimental values were taken from the Pubmed data base, while simulated values were taken from Souza et al. Nat. Methods 2021 (ref 25) or obtained from the simulations presented in this paper.

K denotes the average solubility constant (Eq. 2), which is indeed the same as the partitioning coefficient of the permeant between the membrane and water. This has now been clarified in the Method section on page 25.

The experimental quantity K_{SV} , is a totally different parameter, namely the dynamic quenching constant for calcein. This is now explained on page 23.

* In general, the authors should to go through the manuscript and be more rigorous about mentioning whether presented data in text/figures/supp. info., and the conclusions thereof, are experimental or simulated. This is easy to do for the authors, and it will help to cite the research in this paper in a correcter way.

Reply: We now explicitly indicate in each figure legend whether data is experimental or simulated.

* I don't find in the text how the presented permeability data is calculated from simulations. Was it equation 1 or equation 2?

Reply: We have used Equation 1, the inhomogeneous solubility-diffusion model, to calculate all permeability coefficients from simulations as is stated in the section Methods, subsection Inhomogeneous solubility-diffusion model (page 25). Namely the first paragraph starts with:

"The inhomogeneous solubility-diffusion (ISD) model was used to calculate the permeability coefficient P using Equation 1"

This is also clearly stated in the section Results, second paragraph (page 4):

"... using the inhomogeneous solubility-diffusion model in which the permeation rate is obtained as an integral across the membrane of the ratio between local solubility and friction (see Methods, Eq. 1)"

* In several figures, the data are presented without mentioning the permeant. For instance, Fig. 1F, Fig. 2C, etc. Same for the supporting info.

Reply: All figure captions have been updated to include this information explicitly

* Page 6, top. For hydrophobic permeants, the deviation of the trend of P versus P_{mw} can also be due to barriers in the membrane head group region. A free energy barrier of 1 kBT (not discussed in the text) changes the permeability (P) more than a free energy well of 1 kBT (reflected by $\log P_{mw}$). See e.g. <https://doi.org/10.1021/acs.jctc.7b00039> The authors can mention the influence of free energy barriers for hydrophobic permeants in their text as a possible cause.

Reply: In Figure 1C, we do not record any significant free energy barriers for the permeation of the simulated weak acids, even in case of the most hydrophobic one (Fig. 1C, level IX) it is < 1 kT. If such barriers exist for realistic permeants, they could indeed affect the slope of P versus $\log P_{mw}$ (or $\log P_{ow}$).

* Page 6. "Increasing the length ... increases the hydrophobic thickness of the membrane (Fig. 2B)." This is not visible to me in Fig. 2B?

Reply: Thank you very much for pointing this out. The information is shown in Figure S2. We have added a reference to this figure in the caption of Fig. 2 and the main text.

* Page 9, top. I thought that there would already be a tertiary phase diagram known for the DPPC, DOPC, cholesterol mix (e.g. ref 26). How does the finding compare with previously published phase diagrams? Is it quantitatively the same, or only qualitatively? I also suggest adding some indication on Fig. 4A/B/C where is the coexistence region, e.g. to see the phase diagram 'superimposed' on these graphs, e.g. with some arrows. That would really tie together the message of the phase transition being responsible for the transition. (This is an optional suggestion.)

Reply: Phase diagrams are known for DPPC, DOPC plus cholesterol/ergosterol mixtures (e.g. Davis et al, 2009, Biophys J 96, 521). However, comparison with phase diagrams is not required for the conclusions that we have drawn from our work; we feel that adding the information would complicate Fig. 4A/B/C.

* I am wondering why the choice made to represent permeability ratio's in Fig. 2A/B, 3A/B, etc.? What was there to gain compared to plotting the real values? Maybe the reason can be added to the text.

Reply: We plotted the relative values to make the comparison for the different solutes easier; the permeability coefficients for e.g. formic acid and lactic differ more than an order of magnitude, which complicates plotting in a single graph. The absolute values are given in the figure legends and the supplementary tables.

* Page 21, Fit of the in vitro kinetics: this subsection is dense. Adding information will help to make it reproducible. Please add the Stern-Volmer equation or the specific equation that is fit, such that the reader can relate the equation to the 'fit' parameters and 'fixed' parameters mentioned in this section. Please add examples of the in vitro kinetics (maybe together with their fit) in the Supp. Info. Please add a few notes on the assumptions that are in this fit (e.g. homogeneous distribution inside liposome? membrane thickness known or neglected? etc.) or refer to a paper that lists all the assumptions and limitations. (This last point is valuable to put the results in perspective, because experimental permeabilities can be quite error prone. Computational permeabilities as well, by the way.)

Reply: We now describe the fitting of the in vitro kinetic data in more detail, present the Stern-Volmer

equation and list the assumption of the model (page 23). We included examples of the in vitro kinetics and fits in supplementary Figure S11 and supplementary Table 6.

* Formula's are not rendered correctly in the manuscript, unfortunately, especially pages 20-24. I've not been able to review formula's.

Reply: We apologize for the mistake and have uploaded a pdf version of the manuscript in which the formulas have been rendered correctly.

* Page 11, "... these cavities are filled with ergosterol or cholesterol when sterols are present in the membrane" Can this statement be backed up? E.g. with a radial distribution function between phospholipids and sterols, or with a free volume graph? Otherwise, can it be made clear that this is speculation?

Reply: Our statement may have given the impression that the rare solvent-filled cavities in the case of DPPC bilayer are suddenly occupied by cholesterol molecules when added to the bilayer, which is not what we meant. The cavities are non-existent when sterols are present in the bilayer. This is directly seen from Fig. 4E (zero solvent accessibility for DPPC+15%cholesterol, but non-zero for DPPC below 1 nm). We have rewritten this paragraph (page 12) to be clear on this point.

* Page 17, "when more phases coexist, solutes preferentially permeate through the more fluid parts of the membrane an/or through the region between the coexisting phases." While this claim is very plausible, it is not written in the paper on what observation this claim is based. How was this assessed?

Reply: This claim is based on the difference between the permeability coefficients and the associated free energy barriers from different phases. Thermodynamic macrostates with lower free energy are more probable compared to macrostates with higher free energy. In our work, we show from experiments that more rigid membrane phases have lower permeability coefficients and from simulations that at the same time they pose higher free energy barriers compared to more fluid membrane phases. It naturally follows that in case of phase coexistence the permeation is more probable through the fluid part. We have removed the claim from the paper (page 19, Conclusions section).

* I do not find Eq. 3 back in the cited reference 71 by Luzzati. Can you point me to the correct location in that paper or to the correct citation?

Reply: We apologize for this incorrect citation. The definition of bilayer thickness was originally published in 1960 in *Acta Crystallographica*, but the text is in French, so we wanted to cite the topically closest publication by the same author that includes the definition of hydrophobic thickness usually called Luzzati thickness. This is given on page 215 in the Reference "Luzzati, V., and Husson, F. (1962). The structure of the liquid-crystalline phases of lipid-water systems. *J. Cell Biol.* 12".

* Page 12, "while its permeation is not detectable" Whose permeation? L-lactic acid or formic acid?

Reply: We edited the text to "while the permeation of L-lactic acid is not detectable ...".

* Page 13, "these differences are reflected in". Which differences are meant here? Please rephrase slightly.

Reply: We edited the text to "The differences in partitioning of hydrophilic and hydrophobic solutes are reflected in....".

* Page 13, "we observe that the fluidity of the membranes decreases" Please add a definition and/or explanation what is meant explicitly with 'fluidity'. Also, was this observation done in the current work or in the work that is cited at the end of the sentence? In the latter case, it would be better to write something like "we observed in other work..." If it was observed in this work, the "fluidity" results should be added to the manuscript or Supp. Info.

Reply: The term fluidity is well established in the context of biological membranes as discussed for instance in <https://www.sciencedirect.com/science/article/pii/S0005273604002032>. In this review, section 2.4, the relationship between membrane fluidity and lipid unsaturation is discussed. A structural point of view of membrane fluidity is given in <http://www.sciencedirect.com/science/article/pii/S0005273616300189>, that is, in terms of lipid tail order, which decrease with increasing unsaturation of the acyl chains. We now present this description on page 14 of the revised manuscript.

* Page 14, "We have hypothesized that this difference in protein diffusivity..." This seems to refer to earlier work, please add the citation. If it is a hypothesis in this specific work, the sentence needs to be rephrased somewhat.

Reply: Thank you, references have been added. For protein diffusivity we refer to Bianchi et al Nat Comm (ref. 43) and for solute permeability to Gabba et al Biophys J (ref. 14).

* Page 15, "We now show that..." The citation to ref 56 is strange. Aren't these permeabilities the experimental results of the present paper?

Reply: Correct. We have removed ref. 56.

* Fig. 3C, 4D, 4E have very poor resolution.

Reply: We have added higher resolution figures.

* I thought Lo had a non-capital "o" as subscript.

Reply: We changed to non-capital "o".

* Citation 58 is empty

Reply: Ref 58 has been deleted.

Reviewer #2 (Remarks to the Author):

Frallicciardi and coauthors have studied the permeation of water and of different solutes (weak acids and glycerol) through synthetic membranes, as function of lipid chain length, chain unsaturation and also membrane organization in the presence or cholesterol and ergosterol. One of their motivation is to understand the difference in the permeability of the membrane of yeasts and of mammalian cells. The originality of their work is the combination of experiments - a fluorescence-based assay that monitors the volume changes upon osmotic shock on liposomes using a stopped-flow system and coarse-grained MD simulations (already developed in their BJ paper (ref. 20))- with the last version (3) of the Martini model. With these simulations, it is possible to have details about the free energy profiles across the membrane and the localization of the water molecules in the presence of the solutes.

They have found a relatively good agreement between the experiments and the simulations, which confirms that the Martini force-field correctly models the transfer of these polar components through membranes. They have also shown that the permeation of these solutes depends on the membrane thickness and on lipid tail unsaturation when the membrane is in a Ld (liquid disordered) state. But, they show that the state of the membrane, Ld, versus Lo (liquid ordered) versus gel has a much stronger effect, which is not really surprising. They also confirm that the phase diagrams with ergosterol and cholesterol are different with the same lipids and thus the solute permeation.

Globally, the work is well-performed and the results certainly useful, but not really surprising. I miss the important message that would justify publication in Nature Communications. In addition, the presentation of the results gives the impression that the authors "discover" that the membrane organization, and thus, the addition of sterols, has an impact on permeation. They describe their results first as an effect of sterol or cholesterol, and only next, consider that strong changes in permeation occur because the addition of these sterols lead to different membrane organizations, when this is an obvious reason. They even perform DSC measurements to measure the transition temperature, when the phase diagrams and these properties are probably available in the literature. It would make more sense to compare the permeation of the solutes for the same phase, in the presence of sterol or cholesterol. The authors also completely ignore the seminal work from E. Evans on water

permeability as a function of membrane composition and state (W. Rawicz et al Biophys. J. 94, 4725 (2008); K. Olbrich, et al Biophys. J. 79, 321 (2000)).

Altogether, I think that this paper should be published in a more specialized journal, such as Biophysical journal.

Reply: We agree that differences in the state of the membrane, Ld, versus Lo versus gel have been observed for water and oxygen permeability in model Ld and Lo membranes in both simulations and experiments. However, we now present a comprehensive analysis of solute permeation of membranes in different states, including weak acids and glycerol. We now cite the seminal work of the Evans group (refs 26,33). We also acknowledge that measurements of transition temperatures for similar model membranes are available in the literature (ref28).

The important message of the paper is that we present for the first time a systematic analysis of membrane permeability to a range of solutes in a variety of physiologically important membrane systems. We don't just focus on water, but test a variety of solutes and extract their permeability along with other descriptors of the physico-chemical state of the membrane. We emphasize that we benchmark the entire experimental analysis with molecular dynamics simulations to gain insight into the mechanism of permeation. Previous MD simulations have been limited by mainly looking at the effects of phase state on passive permeation; we analyzed a wide range of lipid conditions. Importantly, although scattered pieces of data are available in the literature, the experimental conditions are almost never the same, making comparisons difficult. We consider the extensive dataset a valuable catalog for future research, whether on the physico-chemical properties of biological membranes or applications in biotechnology (improvement of the strain for productivity) or biomedicine (drug permeability).

We apologize that our writing has given the wrong impression about the effects of sterol. We now say that the effects relate to a different membrane organization, which leads to differences in permeation. Furthermore, we connect the differences observed with cholesterol and ergosterol to the different properties of the plasma membrane of mammalian cells and fungi, respectively (pages 15-17). In brief, by comparing the results of model and yeast plasma membranes, we infer that the difference in permeation lays in the highly rigid and ordered structure of the yeast membrane. Ergosterol is found to modulate membrane permeability and fluidity of highly saturated membranes, but cholesterol does not. This suggests that the yeast membrane has co-evolved lipid saturation and ergosterol, whereas mammalian cells have chosen a different strategy involving the use of cholesterol. These new findings will make our work of interest for the broad readership of *Nature Communications*.

Minor

- In general, the temperature of the experiments should be indicated

Reply: The experiments have been carried out at 20 °C (see page 20), but this information is now also indicated in the figure legends and introduction (page 3).

- Figure 1: Why not also plotting the results of the experiments versus those of the simulations for the same solutes to stress the agreement?

Reply: We like the suggestion. The plot is presented in panel 1G.

- Page 6: the exact nature of the lipids should be detailed in the text (instead of "lipids with mono-unsaturated tails of lengths between 14 and 26 carbons"), instead of in the Supplementary Information

Reply: We now detail the nature of the lipids in the Results section (legend to Figure 2).

- Page 6 "Increasing the length of the phospholipid(Fig. 2B)." It should be Fig. 2C

Reply: Increasing length is shown in Fig. 2B and 2C.

- Figure 2, Legend: the level of unsaturation of the chain and the solute considered for the simulations should be detailed

Reply: Done.

- Page 7: The definition of the degree of unsaturation is not provided and this term is ambiguous. It would be much better to explain that mixtures of saturated and unsaturated lipids have been studied with different respective fractions, and indicate which lipids

Reply: The paragraph describing the degree of unsaturation is now described in the Results section (page 8).

- Fig. S4: the colors do not correspond to the legend

Reply: Corrected.

REVIEWERS' COMMENTS

Reviewer #2 (Remarks to the Author):

The revised version and the answers of the authors to my comments as well as to those of the other referee have convinced me. I now support publication of the new version of the manuscript in Nature Communications.